# Association between Motoric Cognitive Risk Syndrome and Indicators of Reflecting Independent Living among Community-Dwelling Older Adults in Japan: A Cross-Sectional Study

**DOI:** 10.3390/healthcare12181808

**Published:** 2024-09-10

**Authors:** Koji Takimoto, Hideaki Takebayashi, Yoshiyuki Yoshikawa, Hiromi Sasano, Soma Tsujishita, Koji Ikeda

**Affiliations:** 1Department of Rehabilitation, Faculty of Health Sciences, Naragakuen University, 3-15-1 Nakatomigaoka, Nara 631-8524, Japan; y-yoshi@naragakuen-u.jp (Y.Y.); sasano@naragakuen-u.jp (H.S.); koji-ikeda@naragakuen-u.jp (K.I.); 2Department of Rehabilitation, Faculty of Health Sciences, University of Kochi Health Sciences, 2500-2 Otsu, Kochi 781-5103, Japan; takebayashi@ko-ken-k3.ac.jp; 3Department of Physical Therapy, Faculty of Rehabilitation, Kobe International University, 9-1-6 Koyocho-naka, Higashinada-ku, Kobe 658-0032, Japan; tsujishita@kobe-kiu.ac.jp

**Keywords:** motoric cognitive risk syndrome (MCR), frailty, the Questionnaire for Medical Checkup of Old-Old (QMCOO), the Japan Science and Technology Agency Index of Competence (JST-IC), community-dwelling older adults

## Abstract

The purpose of this study was to examine whether motoric cognitive risk syndrome (MCR) is associated with various indicators of independent living among community-dwelling older adults in Japan. The study design was a cross-sectional study, and the participants were 107 community-dwelling older adults (mean age 79 ± 7 years) who were living independently. The participants were administered the Questionnaire for Medical Checkup of Old-Old (QMCOO) as an indicator of health status and the Japan Science and Technology Agency Index of Competence (JST-IC) as an indicator of higher levels of functioning capacity, among others. In addition, we assessed physical frailty (J-CHS), sarcopenia (AWGS2019), and MCR (slow gait + subjective memory complaints), which are predictors of adverse events in the elderly. Multiple regression analysis with QMCOO as the response variable showed that MCR (*p* = 0.01, β: 0.25) and physical frailty (*p* < 0.01, β: 0.43) were significantly associated. In addition, analysis with JST-IC as the response variable showed that MCR (*p* = 0.03, β: −0.20), physical frailty (*p* = 0.01, β: −0.24) and age (*p* = 0.02, β: −0.21) were significantly associated. In conclusion, MCR was found to be similarly associated with QMCOO and JST-IC as physical frailty. It is expected that the MCR will be used as an initial screening tool to identify signs of risk in community-dwelling older people, as it is easy to diagnose.

## 1. Introduction

Motoric cognitive risk syndrome (MCR) is defined as a condition profile that combines a reduced walking speed and subjective memory complaints (SMC). It is a known indicator of an elevated risk of incident dementia [1,2]. A number of meta-analyses and longitudinal studies have demonstrated a correlation between MCR and the incidence of dementia [3,4,5]. Prior research has demonstrated that MCR is linked to diminished cerebral volume [6,7,8] and pathological alterations in the vasculature of cerebral white matter [9]. Additionally, it is associated with lacunar lesions and frontal lobe dysfunction, encompassing the premotor and prefrontal cortices [3,10]. Recently, it has been reported that MCR reflects various other risks besides dementia. For example, MCR has been shown to reflect the risk of falls, disability, and death. Furthermore, these trends have been observed to manifest similarly across racial groups [11,12].

The prevalence of MCR is subject to variation according to the findings of previous studies. Recent meta-analyses report a prevalence of MCR of 10.0% (95 CI: 8.0–12.0%) worldwide [13] and 6.4% [14] and 11.1% [15] in Japan. In Japan, MCR has also been reported to be associated with diabetes, depressive symptoms, falls and disability [14,16]. However, there is a paucity of studies examining the association between MCR and key indicators that are crucial for older people to maintain their independence living in the community.

In recent years, there have been reports indicating that social participation plays a significant role in maintaining the health and well-being of older individuals [17]. A state of social frailty has been identified as a risk factor for physical and cognitive decline, as well as an increased likelihood of developing disabilities [18,19]. Indeed, social frailty has been linked to an elevated risk of incident MCR [20]. It can be reasonably inferred that there may be a correlation between MCR and social participation. Therefore, there may be a link between MCR and social participation. However, the extent to which MCR affects older people’s higher levels of functional living capacity, including aspects of social participation, has yet to be elucidated. It is also known that cognitive and gait functions, which are components of MCR, are closely related to each other [21], and a decline in one function will cause the other to follow suit. This suggests that MCRs may reflect earlier signs that an elderly person’s health is beginning to decline.

In Japan, for example, a questionnaire survey called the Kihon checklist has been used to identify older people who are likely to require long-term care and to implement care prevention programs. However, the government’s estimate that about 5% of the elderly who are not certified as requiring long-term care should be eligible for long-term care prevention projects has not been reached, and the actual figure is less than 1% [22]. Alternatively, the Community Rehabilitation Activity Support Program, which promotes the involvement of rehabilitation professionals to strengthen care prevention efforts in the community, was launched in 2012. However, 65.2% of all municipalities in Japan have dispatched physiotherapists and 46.8% of occupational therapists, leaving a shortage of professionals who can contribute to care prevention in the community [23]. In addition, the Questionnaire for Medical Checkup of the Old-Old (QMCOO) has been adopted from 2020 [24], but its ability to predict incident disability among the elderly has yet to be verified [25]. Therefore, in Japan, a super-aged society, there is a need for a simple disability prediction tool to complement the Kihon checklist and the QMCOO.

The present study was designed to test the hypothesis that MCR is an indicator that reflects independent living indicators such as social participation and the health status of the elderly. To test this hypothesis, an MCR assessment was conducted on community-dwelling older adults as well as a comprehensive evaluation of health status and social participation, and the association between these factors and MCR was examined. Furthermore, the extent to which MCR explains the health status and social participation of the elderly, independently of the conventionally used measures of physical frailty and sarcopenia, was examined.

## 2. Methods

### 2.1. Study Design

A cross-sectional study was conducted between June 2023 and June 2024 among community-dwelling elderly persons. The study was approved by the Nara Gakuen University Ethics Committee (4-H008). Informed consent was obtained from all individuals who participated in the study before the study was conducted.

### 2.2. Calculation of Sample Size

Sample sizes were calculated using G*Power 3.1.9.7 (Heinrich Heine University, Düsseldorf, Germany). In a multiple regression model with five explanatory variables, calculated as effect size 0.15 (medium), power 80%, and alpha error 0.05, the total sample size was 92. The number of participants was set at 140, considering the possibility of a reduction in the number of participants due to the exclusion criteria, as well as the prevalence of MCR (about 10%) [13].

### 2.3. Participants

The participants of this study were community-dwelling older adults aged 65 years or older living independently in Nara city (Nara prefecture), who participated in a resident-led exercise class (self-management exercise group) being run in Nara city. Self-management exercise groups are known to be effective in preventing disability [26,27] and are widely practiced throughout Japan. Self-management exercise groups meet on a weekly to fortnightly basis, with group exercises (exercise programs consisting of stretching, resistance training, etc.) lasting 60–90 min per session. The subjects of the study were those who responded to a request for research cooperation made to participants of a self-management exercise group through the Community Comprehensive Support Center in Nara city. Exclusion criteria for participants were as follows: they did not have dementia (never diagnosed by a doctor), were independent in their daily lives, were certified by the long-term care insurance system as requiring long-term care (level of care 1 or above), had a history of central nervous system disease, neurocognitive disorder or psychiatric illness, and were currently undergoing treatment and attending hospital. A total of 3 of the 140 participants met the exclusion criteria. In addition, 30 participants had missing values on physical function tests or questionnaire responses. Finally, data from 107 participants were included in the statistical analysis (Figure 1).

### 2.4. Questionnaire Survey and Measurement Methods

In this study, five outcome measures (QMCOO, JST-IC, social frailty, LSNS-6, and LSA; see below for explanation) reflecting health status and various life abilities for independent living were set as objective variables, and physical frailty and sarcopenia, known risk indicators for the elderly, were used as explanatory variables along with MCR to examine and measure the following factors.

#### 2.4.1. Basic Information on Participants

Basic information on the participants was obtained by means of a questionnaire, including age, gender, years of education and whether they were certified as requiring care under the long-term care insurance system.

#### 2.4.2. MCR Criteria

The criteria for MCR followed the definition by Verghese et al. [1] of having both subjective memory complaints (SMC) and slow gait (SG), being independent in activities of daily living and having no dementia. The SMC was determined using one of the questions from the 15-item Geriatric Depression Scale (GDS): ‘Do you feel you have more problems with memory than most?’. If the answer to this question was ‘yes’, the participant was classified as having SMC [28]. This method of SMC determination has been used in 7 of the 17 studies included in the MCR prevalence study [13]. Walking speed was measured on a flat 5 m walking path at a usual pace, with a 2 m spare path at each end of the walking path as an acceleration and deceleration section. Participants were instructed to walk a 5 m walking section at their usual pace and the measurer measured the time required with a stopwatch. The time required to walk 5 m was converted into speed (m/s) and was considered as SG if it was less than the mean value of the participants in this study minus 1 standard deviation (SD). In previous studies, the mean and SD were calculated by age and sex and SD determination was performed [14], but in this study, due to the limited number of participants, the reference values were calculated from the mean and standard deviation of all participants. Those who fell under both SMC and SG were judged as having MCR. Note that being independent in daily living and not having dementia were judged on the basis that the participants did not fall into the categories of those requiring long-term care certification under the long-term care insurance system.

#### 2.4.3. The 15-Item Questionnaire for Medical Checkup of Old-Old (QMCOO)

In Japan, the QMCOO, a questionnaire suitable for assessing the health status of people aged 75 years and older, has been used since 2020 [25]. The QMCOO is used to assess the multidimensional frailty of older people and consists of 15 questions within 10 domains: ‘health status, mental health, eating behavior, oral function, weight loss, physical function and falls, cognitive function, smoking, social participation and social support’ [24]. The response format is a choice-response method, mainly based on two options of ‘yes’ and ‘no’.

The QMCOO is known to effectively determine frailty, with higher scores (range 0–15 points) reflecting a more advanced degree of frailty. According to previous studies using QMCOO to diagnose frailty, a cutoff of 3/4 points can be used to determine frailty and a cutoff of 2/3 points can be used to determine pre-frailty [29,30]. This study used the QMCOO total score.

#### 2.4.4. The Japan Science and Technology Agency Index of Competence (JST-IC)

The JST-IC is a questionnaire that can effectively assess the health status and social inactivity of older people and is used as a tool for earlier care and isolation prevention for community-dwelling older adults [31,32]. The JST-IC is characterized by the fact that it assesses the higher levels of functional living capacity required for older people to lead independent and active daily lives in the community, rather than just being independent in basic ADLs. The JST-IC consists of 16 questions covering four areas: the ability to use new equipment, the ability to gather and use information necessary for a better life, the ability to manage one’s own life and the lives of those around them, including family members, and the ability to participate in local activities and play a role in the community [32]. The questionnaire was answered using a ‘yes’ or ‘no’ response, with higher scores (range 0–16 points) reflecting more advanced levels of functional living capacity. The JST-IC was also scored and used in this study.

#### 2.4.5. Social Frailty Criteria

To assess the social frailty of community-dwelling older adults, the five-question criterion by Makizako et al. was used [18,19]. Social frailty by this assessment method is known to be associated with cognitive and physical functioning decline [33]. The questionnaire items were (1) going out less frequently, (2) rarely visiting friends, (3) feeling unhelpful to friends or family, (4) living alone and (5) not talking with someone every day. Cases with no indicators were judged as robust, cases with only one were judged as socially prefrail and cases with two or more were judged as socially frail.

#### 2.4.6. The Lubben Social Network Scale-6 (LSNS-6)

The Lubben Social Network Scale is a questionnaire used to measure the social networks of older people [34], and a shortened version of the LSNS-6 [35] with only six questions was used in this study. The LSNS-6 questionnaire consists of six items, three related to family networks and three related to non-family (e.g., friends) networks. Each item is answered by indicating the number of people in the network using a six-point scale. The total score is 30 points, with higher scores interpreted as signifying greater social networks.

#### 2.4.7. Life Space Assessment (LSA)

LSA is an indicator that assesses the mobility of an individual’s living space and their usual mobility patterns over the last month [36]. The questionnaire assesses the frequency of travel and mobility independence for each of five levels of living space: the bedroom, the home, outside, the neighborhood, the town and unlimited space. The total score is 120, with higher scores reflecting more living space.

#### 2.4.8. Physical Frailty Criteria

The criteria used to assess physical frailty were the J-CHS criteria [37], based on the CHS criteria [38]. The J-CHS was designed to assess whether the individual had (1) unintentionally lost 2 or more kg in the past 6 months (yes), (2) grip strength (<28 kg in men or <18 kg in women), (3) felt tired in the past 2 weeks without a reason (yes), (4) gait speed (<1.0 m/s), (5) engaged in moderate physical exercise or sports aimed at health or engaged in low levels of physical exercise aimed at health (all no). Cases with no indicators were judged as robust, cases with 1~2 were judged as prefrail and cases with 3 or more were judged as frail.

#### 2.4.9. Sarcopenia Criteria

The Asian Working Group for Sarcopenia’s 2019 clinical research settings were used to determine sarcopenia [39]. Grip strength (men < 28 kg, women < 18 kg) as a measure of muscle strength, gait speed (<1.0 m/s) as a measure of physical performance and bioelectrical impedance analysis (<7.0 kg/m^2^ in men, <5.7 kg/m^2^ in women) as a measure of skeletal muscle mass index (SMI) were employed. SMI was measured using an InBody 470 (InBody Japan, Tokyo, Japan). Sarcopenia was diagnosed when low skeletal muscle mass was accompanied by either reduced grip strength or reduced gait speed, and severe sarcopenia was diagnosed when all the above were applicable.

### 2.5. Statistical Analysis

To check for differences between the MCR and non-MCR groups, an unpaired *t*-test or Mann–Whitney U test was performed on age, BMI and other outcome measures. The chi-square test or Fisher’s exact test was also conducted to confirm the association between MCR presence and gender, physical frailty, social frailty and sarcopenia. Similarly, one-way analysis of variance and post hoc tests (Tukey’s method) were conducted for differences in outcome measures based on physical frailty and sarcopenia determination. Outcome measures that were found to be significantly different or associations between those with and without MCR were then taken as the objective variables. Multivariate analyses (multiple regression analysis or logistic regression analysis) were then conducted using MCR, physical frailty and sarcopenia as explanatory variables, and age and gender as confounding factors. For all statistical analyses, the significance level was set at 5%. The software used for statistical analysis was BellCurve for Excel Ver 4.07 and R Ver 4.0.2 for Windows.

## 3. Results

### 3.1. Characteristics of the Participants

The characteristics of the participants are shown in Table 1. The cut-off value for gait speed for MCR determination was set at 0.85 m/sec, which is the mean −1 SD value of the participants. Results showed a prevalence of MCR of 13/107 (12.1%). Significant differences were found between MCR and non-MCR in QMCOO (*p* < 0.001) and JST-IC (*p* < 0.001) as well as age (*p* < 0.001). Regarding the association between MCR, frailty and sarcopenia determination, a significant association was found for both social frailty (*p* = 0.021), physical frailty (*p* < 0.001) and sarcopenia (*p* = 0.002).

### 3.2. Results of Outcome Measures by Physical Frailty Determination

The J-CHS physical frailty assessment showed that the prevalence of each was 15.9% (n = 17) frail, 44.9% (n = 48) prefrail and 39.3% (n = 42) robust. When examining whether there were differences or associations in outcome measures by physical frailty determination outcome, the main effects and associations were found for all outcome measures (Table 2).

### 3.3. Results of Outcome Measures by Sarcopenia Determination

The results of the sarcopenia assessment showed that the prevalence of each was 8.4% (n = 9) for severe sarcopenia, 7.5% (n = 8) for sarcopenia and 84.1% (n = 90) for robust. When examining whether there were differences or associations in outcome measures by sarcopenia determination outcome, the main effects and associations were found for all outcome measures (Table 3).

### 3.4. Results of Multivariate Analysis

The outcome measures QMCOO, JST-IC and social frailty, which were significantly different or associated between the MCR and non-MCR groups, were selected as the objective variables for the multivariate analysis.
First multivariate analysis: multiple regression analysis with QMCOO as the objective variable resulted in a coefficient of determination (R^2^) of 0.33, the regression equation was significant (*p* < 0.001) and MCR (β = 0.25, *p* = 0.005) and physical frailty (β = 0.43, *p* < 0.001) were selected as significant explanatory variables (Table 4).


Second multivariate analysis: multiple regression analysis with JST-IC as the objective variable showed that the coefficient of determination (R^2^) was 0.37, the regression equation was significant (*p* < 0.01) and the significant explanatory variables MCR (β = −0.20, *p* = 0.028), sarcopenia (β = −0.18, *p* = 0.060), physical frailty (β = −0.24, *p* = 0.011) and age (β = −0.21, *p* = 0.020 were selected (Table 5).



Third multivariate analysis: binomial logistic regression analysis with social frailty as the objective variable (broadly divided into two groups according to presence or absence of social frailty) showed that the coefficient of determination (R^2^) was 0.13, the regression equation was significant (*p* < 0.001) and MCR (OR: 3.14, *p* = 0.107), physical frailty (OR = 2.15, *p* = 0.029) and sex (OR = 0.26, *p* = 0.010) were selected as variables to be included in the regression equation (Table 6).


## 4. Discussion

This study aimed to test the hypothesis that MCR is associated with various health indicators in community-dwelling older adults. Furthermore, the extent to which MCR was associated with the health indicators utilized in this study was examined in comparison to the established risk indicators of physical frailty and sarcopenia. The main findings indicate that MCR is associated with the health status (QMCOO) and higher life functioning (JST-IC) of community-dwelling older adults. Moreover, MCR was identified as a reflective indicator of the health status and higher life functioning of older individuals, exhibiting similarities to physical frailty.

The prevalence of MCR in the participants of this study was 12.1%, which is generally comparable to the MCR prevalence in previous studies [13] (range 4.22–14.74). The cut-off value of gait speed for determining MCR in this study was 0.85 m/s, which is the mean −1 SD value of the participants. The mean gait speed of the MCR participants was 0.75 ± 0.09 m/s. This cut-off value for gait speed is comparable to those used in other studies on Japanese subjects [14,15,16], and the results of the present MCR determination appear to be generally valid.

One of the findings of the present study was the discovery of an association between QMCOO and MCR. The QMCOO is designed to encompass a comprehensive range of frailty conditions, many of which may be associated with physical frailty. Indeed, the possibility of physical frailty determination has been investigated by QMCOO and it has been reported that frailty determination is possible [29,40]. The QMCOO questions include ‘7: Do you think you walk slower than before?’, which asks about subjective gait speed reduction. In addition, there are questions such as ‘10: Do your family or your friends point out your memory loss? For example, you ask the same question over and over again’ or ‘11: Do you find yourself not knowing today’s date?’, which are questions about cognitive decline, similar to the MCR judgement. This feature of the QMCOO structure is the reason why it was found to be associated with MCR. A recent study reported that subjective MCRs reflect the original MCR [41], which also supports the interpretation of the present findings.

On the other hand, the association between JST-IC and MCR is another interesting finding of this investigation. The JST-IC encompasses a range of higher levels of functional living capacity that are essential for older adults to maintain an independent lifestyle. The JST-IC questionnaire encompasses competencies that necessitate advanced cognitive functioning, including information literacy and the utilization of novel equipment. With regard to cognitive functioning, there have been reports indicating that SMC is associated with health literacy [42] and that the capacity to utilize everyday technology begins to decline at the SMC stage [43,44]. In this context, it is noteworthy that higher levels of functional living capacity in older individuals necessitate the maintenance of a high level of cognitive function and that MCR was associated with indicators of older individuals’ ability to live independently.

The final indicator, social frailty, showed no correlation with MCR. However, a longitudinal study examining the association between social frailty and the incidence of MCR [20] has shown that individuals with social frailty are at an elevated risk of developing MCR compared to those without social frailty. The fact that this study used a cross-sectional design and the restriction of the participants to a single city may have contributed to the absence of an association between MCR and social frailty. It should be noted that the QMCOO and JST-IC include questions pertaining to social participation. In the QMCOO, three of the total fifteen questions are related to social participation and social support, whereas in the JST-IC, four of the total sixteen questions are concerned with social participation. It has been reported that social participation can reduce the risk of physical functional decline [17,45] and reduce the risk of developing a need for long-term care [17] in community-dwelling older adults. The findings of the present study indicate that MCRs do not fully account for social participation status. Therefore, a separate assessment of the social participation of community-dwelling older adults needs to be carried out.

Our findings suggest that MCRs may enable community-dwelling older adults to assess their higher levels of functional living capacity and health status, which are important factors in independent living, through a simple and easy-to-assess MCR. Furthermore, SMC and SG, which are components of the MCR, have been demonstrated to reflect a range of adverse events when assessed individually [46,47,48]. Therefore, the MCR should be used for screening community-dwelling older adults because, in addition to being a simple assessment, it reflects the comprehensive health status and higher life functions of older people. However, there are challenges in using the MCR as a replacement for conventional assessment tools such as frailty and sarcopenia. More careful assessment is needed to identify in detail the problems of older people with varying degrees of frailty. In such cases, surveys and medical examinations using QMCOO and the Kihon checklist [49] are considered necessary. However, as the MCR assessment does not require any special expertise, it can be used to help community-based long-term care prevention projects to identify the target population. Alternatively, it could be used to improve health awareness by having residents conduct MCR assessments among themselves.

The limitations of this study should be noted. First, the study is a cross-sectional study and cannot explain the causal relationship between MCR and community-dwelling older adults’ health status (QMCOO) and higher life functioning (JST-IC). Secondly, the study’s participants were older people living in a particular area, so the results may reflect effects specific to the area of interest. Third, the participants in this study participated in a self-management exercise group, and the influence of this activity may be reflected in the results. Fourth, there are many other factors that can influence health status and higher life functions than those addressed here, which have not been taken into account (e.g., built environment, objective medical information, etc.). In addition, global cognitive function tests such as the MMSE were not available for the subjects in this study. In recent years, it has become known that the presence of negative affect as well as depressive symptoms in older people can have a negative impact on independent living and daily functioning [50,51]. It is also known worldwide that the prevalence of frailty is higher in women. There are interesting findings that the higher prevalence of frailty in women is related to financial exploitation [52], and we believe that it is important to discuss this issue in the future, taking into account different cultural backgrounds and gender and their impact on independent living among the elderly.

Future research should follow participants longitudinally, complementing factors affecting health status and higher life functioning, to demonstrate the effectiveness of conducting MCR determinations for community-dwelling older adults.

## 5. Conclusions

This study examined whether MCR is associated with indicators of independent living among community-dwelling older adults. In conclusion, MCR was found to be similarly associated with health status (QMCOO) and higher levels of functional living capacity (JST-IC) as physical frailty. It is expected that the MCR will be used as an initial screening tool to identify signs of risk in community-dwelling older people, as it is easy to diagnose.

## Figures and Tables

**Figure 1 healthcare-12-01808-f001:**
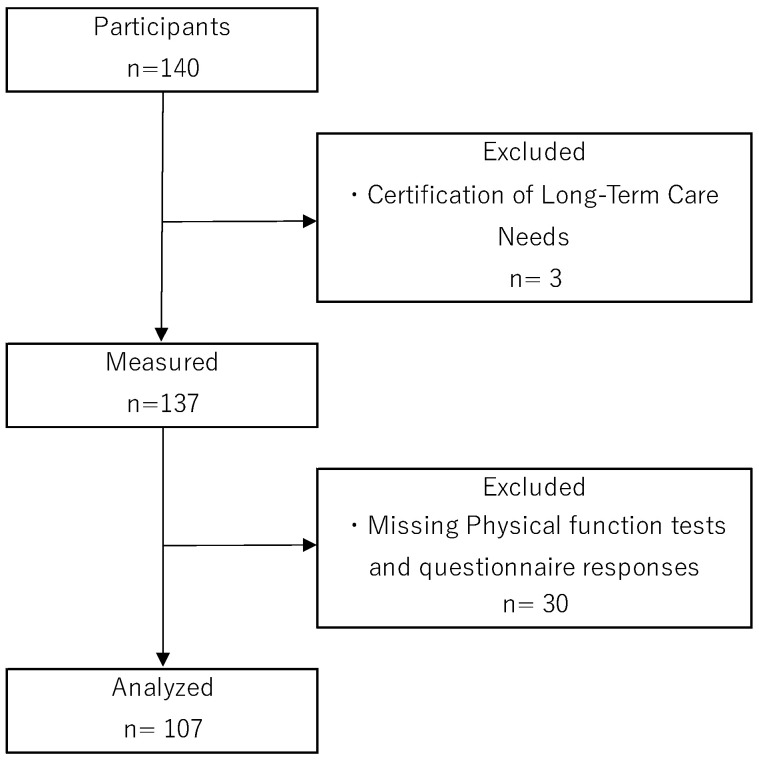
Flowchart of participant selection.

**Table 1 healthcare-12-01808-t001:** Characteristics of the participants and comparison of MCR and non-MCR.

	MCR	non-MCR	*p*-value	effect size	
	(n = 13, 12.1%)	(n = 94, 87.9%)			
Age (years)	86.6 ± 5.1	78.3 ± 6.4	<0.001	0.40	^#^
Sex, women (n, %)	10 (76.9)	73 (77.7)	1.000	0.01	**
BMI (kg/m^2^)	21.2 ± 3.3	22.6 ± 3.0	0.131	0.45	^#^
Education (years)	12.5 ± 2.3	13.3 ± 2.4	0.211	0.35	^###^
Fall (n, %)	3 (23.1)	13 (13.8)	0.408	0.08	**
Gait Speed (m/s)	0.7 ± 0.1	1.2 ± 0.2	<0.001	0.88	^##^
SMC (n, %)	13 (100)	44 (46.8)	<0.001	0.35	*
QMCOO (point)	4.2 ± 1.5	2.0 ± 1.6	<0.001	0.43	^###^
JST-IC (point)	7.8 ± 4.2	12.7 ± 3.0	<0.001	0.45	^###^
LSNS-6 (point)	14.5 ± 7.3	15.4 ± 5.5	0.565	0.06	^#^
LSA (point)	71.7 ± 28.1	86.0 ± 22.8	0.113	0.20	^###^
Social Frailty (n, %)			0.021	0.27	**
Robust	1 (7.7)	35 (37.2)			
Social Pre-Frail	3 (23.1)	30 (31.9)			
Social Frail	9 (69.2)	29 (30.9)			
Physical Frailty (n, %)			<0.001	0.42	**
Robust	0 (0.0)	42 (44.7)			
Pre-Frail	6 (46.2)	42 (44.7)			
Frail	7 (53.8)	10 (10.6)			
Sarcopenia (n, %)			0.002	0.40	**
Robust	7 (53.8)	83 (88.3)			
Sarcopenia	1 (7.7)	7 (7.4)			
Severe-Sarcopenia	5 (38.5)	4 (6.3)			

Values are mean ± SD or n and proportion. #: unpaired *t*-test, ##: Welch’s *t*-test, ###: Mann–Whitney U test, *: chi-square test, **: Fisher’s exact test. Effect size: chi-square test is phi coefficient, Fisher’s exact test is Cramer’s V, others: Cohen’s d. BMI: Body Mass Index, SMC: Subjective Memory Complaints, QMCOO: Questionnaire for Medical Checkup of Old-Old, JST-IC: Japan Science and Technology Agency Index of Competence, LSNS-6: Lubben Social Network Scale-6, LSA: Life Space Assessment.

**Table 2 healthcare-12-01808-t002:** Comparison of groups classified by physical frailty.

	Robust ^a^	Pre-Frail ^b^	Frail ^c^	*p*-value	η^2^	post-hoc test
	(n = 42)	(n = 48)	(n = 17)			
QMCOO (point)	1.2 ± 1.0	2.7 ± 1.6	3.7 ± 2.1	<0.001	0.71	a < b < c
JST-IC (point)	13.3 ± 2.8	12.6 ± 2.7	7.6 ± 3.8	<0.001	0.68	c < a, b
LSNS-6 (point)	16.3 ± 5.7	15.8 ± 5.1	11.6 ± 6.0	0.012	0.92	c < a, b
LSA (point)	90.3 ± 24.3	85.4 ± 19.7	66.4 ± 25.5	0.002	0.88	c < a, b
Social Frailty (n, %)				*p*-value	Cramer’s V	
Robust	19 (52.8)	13 (36.1)	4 (11.1)	0.024	0.229	
Social Pre-Frail	13 (39.4)	18 (54.5)	2 (6.1)			
Social Frail	10 (26.3)	17 (44.7)	11 (28.9)			

Values are mean ± SD or n and proportion. QMCOO, JST-IC, LSNS-6, LSA: one-way ANOVA and post hoc test (Tukey’s method), Social Frailty: chi-square test. Effect size: one-way ANOVA is η^2^, chi-square test is Cramer’s V. a (Robust), b (Pre-Frail), and c (Frail) are reflected in the results of the post-hoc test. QMCOO: Questionnaire for Medical Checkup of Old-Old, JST-IC: Japan Science and Technology Agency Index of Competence, LSNS-6: Lubben Social Network Scale-6, LSA: Life Space Assessment.

**Table 3 healthcare-12-01808-t003:** Comparison of groups classified by sarcopenia.

	Robust ^a^	Sarcopenia ^b^	Severe Sarcopenia ^c^	*p*-value	η^2^	post-hoc test
	(n = 90)	(n = 8)	(n = 9)			
QMCOO (point)	1.9 ± 1.6	3.6 ± 1.3	4.0 ± 2.0	<0.001	0.84	a < b, c
JST-IC (point)	12.8 ± 3.1	9.0 ± 3.1	7.7 ± 2.9	<0.001	0.77	b, c < a
LSNS-6 (point)	15.9 ± 5.4	11.0 ± 5.9	13.0 ± 6.6	0.027	0.93	b < a
LSA (point)	86.5 ± 22.3	78.8 ± 32.1	66.8 ± 25.1	0.046	0.94	c < a
Social Frailty (n, %)				*p*-value	Cramer’s V	
Robust	34 (94.4)	0 (0.0)	2 (5.6)	0.014	0.237	
Social Pre-Frail	30 (90.9)	2 (6.1)	1 (3.0)			
Social Frail	26 (68.4)	6 (15.8)	6 (15.8)			

Values are mean ± SD or n and proportion. QMCOO, JST-IC, LSNS-6, LSA: one-way ANOVA and post hoc test (Tukey’s method), Social Frailty: Fisher’s exact test. Effect size: one-way ANOVA is η^2^, Fisher’s exact test is Cramer’s V. a (Robust), b (Sarcopenia), and c (Severe Sarcopenia) are reflected in the results of the post-hoc test. QMCOO: Questionnaire for Medical Checkup of Old-Old, JST-IC: Japan Science and Technology Agency Index of Competence, LSNS-6: Lubben Social Network Scale-6, LSA: Life Space Assessment.

**Table 4 healthcare-12-01808-t004:** Association of QMCOO with MCR and other variables.

			95%CI		
	B	β	min	max	*p*-value
MCR	1.32	0.25	0.41	2.24	0.005
Physical Frailty	1.07	0.43	0.64	1.49	<0.001

Adjusted R^2^: 0.33, objective variable: QMCOO, explanatory variables: MCR, physical frailty, sarcopenia, age, sex. CI: confidence interval, MCR: motoric cognitive risk syndrome.

**Table 5 healthcare-12-01808-t005:** Association of JST-IC with MCR and other variables.

			95%CI		
	B	β	min	max	*p*-value
MCR	−2.14	−0.20	−4.04	−0.23	0.028
Sarcopenia	−1.07	−0.18	−2.18	0.05	0.060
Physical Frailty	−1.19	−0.24	−2.11	−0.28	0.011
age	−0.11	−0.21	−0.20	−0.02	0.020

Adjusted R^2^: 0.37, objective variable: JST-IC, explanatory variables: MCR, physical frailty, sarcopenia, age, sex. CI: confidence interval, MCR: motoric cognitive risk syndrome.

**Table 6 healthcare-12-01808-t006:** Association of social frailty with MCR and other variables.

		95%CI		
	OR	min	max	*p*-value
MCR	3.14	0.78	12.66	0.107
Physical Frailty	2.15	1.08	4.28	0.029
Sex	0.26	0.09	0.72	0.010

Adjusted R^2^: 0.13, objective variable: social frailty/non-frailty, explanatory variables: MCR, physical frailty, sarcopenia, age, sex. OR: odds ratio, CI: confidence interval, MCR: motoric cognitive risk syndrome.

## Data Availability

The data supporting the findings of this study are available upon request from the corresponding author. The data are not publicly available because of privacy and ethical restrictions.

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
