# Peer review of "Association between Motoric Cognitive Risk Syndrome and Indicators of Reflecting Independent Living among Community-Dwelling Older Adults in Japan: A Cross-Sectional Study"

_healthcare, 2024, doi:10.3390/healthcare12181808_

Round 1
Reviewer 1 Report
Comments and Suggestions for Authors
Dear Authors,
Thank you for the valuable research.
1- Please, explain the QMCOO clearly. Your description of this questionnaire is not clear. " The QMCOO is known to effectively determine frailty, with higher scores (range 0-15 points) reflecting a more advanced degree of frailty, and 4 points or more effectively determining frailty [27]." Please write the cutoff point of QMCOO. I found this study, please use it : "Validation of the questionnaire for medical checkup of old-old (QMCOO) score cutoff to diagnose frailty" Mitsutaka Yakabe, Koji Shibasaki, Tatsuya Hosoi, Shoya Matsumoto, Kazuhiro Hoshi, Masahiro Akishita & Sumito Ogawa; BMC Geriatrics volume 23, Article number: 157 (2023)
2- Please mention the opinions of your research team about the use of QMCOO in the elderly with different degrees of frailty in the final part of the discussion and conclusion.
For instance, considering your findings, is it possible for this tool to be substituted by other tools that have established a significant connection with QMCOO?
Or in other words, do you consider the use of this tool as a priority over other checkup for older adult (mentioned in this research)?
3- Some parts of the manuscript (introduction and discussion) have been written using artificial intelligence. Please humanize it.
Best regards,
Author Response
We sincerely thank you for your thought-provoking advice on our submitted paper, “Association between Motoric Cognitive Risk Syndrome and Indicators of Reflecting Independent Living among Community-Dwelling Older Adults in Japan: A Cross-Sectional Study”.
We have responded to your comments as follows, which you can review along with the revised manuscript.
Comment 1: Please, explain the QMCOO clearly. Your description of this questionnaire is not clear. " The QMCOO is known to effectively determine frailty, with higher scores (range 0-15 points) reflecting a more advanced degree of frailty, and 4 points or more effectively determining frailty [27]." Please write the cutoff point of QMCOO. I found this study, please use it : "Validation of the questionnaire for medical checkup of old-old (QMCOO) score cutoff to diagnose frailty" Mitsutaka Yakabe, Koji Shibasaki, Tatsuya Hosoi, Shoya Matsumoto, Kazuhiro Hoshi, Masahiro Akishita & Sumito Ogawa; BMC Geriatrics volume 23, Article number: 157 (2023)
Response 1: Thank you for your comments on QMCOO, we have cited the paper you provided on P4, L180-182 and also clarified the cut-off point of QMCOO. Please note that in this study, only the total score of QMCOO was used as the index of consideration, and the cut-off point was not used to determine frailty. Therefore, we have limited our explanation of QMCOO to this extent.
Comment 2: Please mention the opinions of your research team about the use of QMCOO in the elderly with different degrees of frailty in the final part of the discussion and conclusion.
For instance, considering your findings, is it possible for this tool to be substituted by other tools that have established a significant connection with QMCOO?
Or in other words, do you consider the use of this tool as a priority over other checkup for older adult (mentioned in this research)?
Response 2: Thank you for your valuable input. We do not believe that MCR is necessarily a substitute for frailty or sarcopenia. We believe that other assessments, including QMCOO, are needed to more carefully identify problems in older adults with varying degrees of frailty. However, the MCR is easier to determine than other tools. For example, it may be possible for community residents to conduct MCR assessments among themselves as an initial screening to help raise health awareness. In addition, because the MCR assessment does not require special expertise, it can be used by municipal care prevention project staff to identify eligible persons. In conclusion, although it is difficult to replace other assessment tools, we believe that the MCR assessment is very useful in that it can be used more easily as an initial screening tool. These points have been added to P9-10, L461-468 of the Discussion and P10, L493-495 of the Conclusion.
Comment 3: Some parts of the manuscript (introduction and discussion) have been written using artificial intelligence. Please humanize it.
Response 3: Thank you for your comments regarding the English text of the Introduction and Discussion. Although the author's English is limited, please confirm that I have reviewed the English text again regarding the Introduction and Discussion you mentioned.
Reviewer 2 Report
Comments and Suggestions for Authors
Although this is an interesting topic, authors need to check the following points:
Please provide in the abstract a more extended conclusion.
How was the MCR diagnosis given? It is not clear.
Authors can also discuss the importance of frailty on independent living and social functioning in older adults coming from other cultural settings (for a relevant recent study to discuss: https://link.springer.com/article/10.14283/jfa.2022.57). The issue of gender should be also discussed.
Why do authors believe that there is an interest in such a research study in Japan? Is there a particular reason? Please justify.
The aim is not supported by relevant literature. Please provide more references.
Please explain the term dementia. Why is not the term neurocognitive disorder used in the exclusion criteria?
Was depression and anxiety levels checked? It has been found that not only depressive symptomatology, but also the existence of negative affect has a detrimental influence on different variables of independent living and everyday functioning in older patients with cognitive deficits as well as in community-dwelling older adults. In the introduction section of the following article authors can find a literature review on the role of depression on cognition, IADLs, but also new findings on the role of negative affect on IADLs in community-dwelling older adults with no depression: https://doi.org/10.1016/j.jadr.2022.100391).
Please describe in more detail the recruitment procedure.
Was MMSE administered? If no, why? Please justify as most studies use this measure for the assessment of global cognitive functioning.
Effect sizes are missing. Please provide them for all applied statistics.
Please follow APA guidelines for presenting statistics in the main text.
There are too many tables. Authors should think about using figures.
The Conclusions section is very brief. Please provide a more detailed section.
Comments on the Quality of English LanguageModerate English language editing throughout the text.
Author Response
We sincerely thank you for your thought-provoking advice on our submitted paper, “Association between Motoric Cognitive Risk Syndrome and Indicators of Reflecting Independent Living among Community-Dwelling Older Adults in Japan: A Cross-Sectional Study”.
We have responded to your comments as follows, which you can review along with the revised manuscript.
Comment 1: Please provide in the abstract a more extended conclusion.
Response 1: Thank you for your remarks about the abstract. Please check P1, L27-29 as we have added and revised the conclusion section of the abstract.
Comment 2: How was the MCR diagnosis given? It is not clear.
Response 2: Thank you for pointing this out the MCR diagnosis. The diagnostic criteria for MCR were determined based on the definition by Verghese et al. [1], as described in Methodology 2.4.2 MCR Criteria (P4, L150-169). The MCR is a condition profile that combines SMC and slow gait, and also requires that the patient is not suffering from dementia and is independent in daily living. First, SMC was determined by answering the GDS (Geriatric Depression Scale) question, “Do you feel that you have more problems with memory than others?’ was judged as SMC if the answer to this question was 'yes'. Slow gait was then determined if the participant's gait speed was 1 SD below the mean of the participants' gait speeds in this study. The cutoff for slow gait in this study was 0.85 m/sec. (P6, L260-262 in Result 3.1). The subjects in this study were elderly people who had not been diagnosed with dementia and were independent in their daily lives. This was determined by the fact that they were not certified as requiring long-term care under the Japanese long-term care insurance system. These are the criteria for MCR diagnosis.
Comment 3: Authors can also discuss the importance of frailty on independent living and social functioning in older adults coming from other cultural settings (for a relevant recent study to discuss: https://link.springer.com/article/10.14283/jfa.2022.57). The issue of gender should be also discussed.
Response 3: Thank you for your very interesting points and findings. Gender differences in the prevalence of frailty are well known around the world, but it is interesting to report that they affect aspects such as financial exploitation. We also think it is important to discuss this in light of the impact of different cultural backgrounds and gender on independent living among the elderly. Please see the paper you referred to in this regard, which we have cited and added (P10, L481-485).
Comment 4: Why do authors believe that there is an interest in such a research study in Japan? Is there a particular reason? Please justify.
Response 4: Thank you for your very important question about our research. In Japan, survey research such as this study is attracting a great deal of attention. In order to implement a more effective high-risk approach in Japan's hyper-aged society, it is necessary to select the target population efficiently. Questionnaires are often used for initial screening of community-dwelling elderly people. However, there is a problem that questionnaire surveys cannot adequately select the target population, and tools are needed to compensate for this, and we believe that the MCR discussed in this study will play an important role as one such tool. Please see the additional information on these points in the Introduction (P2, L65-78).
Comment 5: The aim is not supported by relevant literature. Please provide more references.
Response 5: The MCR used in this study is relevant to the purpose of this study because it is needed as a simple assessment tool that reflects the life functions of the elderly. The Kihon checklist and QMCOO, which have been used in Japan, are inadequate tools for predicting disability. However, if the MCR, which can be easily determined, is shown to be related to the health status and other factors of community-dwelling elderly, it is expected to be an effective disability prediction tool that complements the Kihon checklist and QMCOO. Please see the addition to the Introduction (P2, L65-78) along with the content of Response 4.
Comment 6: Please explain the term dementia. Why is not the term neurocognitive disorder used in the exclusion criteria?
Response 6: Thank you for your comments about dementia. In this study, dementia refers to a physician-diagnosed decline in cognitive function. Although the term neurocognitive impairment is not used in the exclusion criteria, we have included it because, in practice, we treat those who fall into this category as excluded (P3, L130, 132).
Comment 7: Was depression and anxiety levels checked? It has been found that not only depressive symptomatology, but also the existence of negative affect has a detrimental influence on different variables of independent living and everyday functioning in older patients with cognitive deficits as well as in community-dwelling older adults. In the introduction section of the following article authors can find a literature review on the role of depression on cognition, IADLs, but also new findings on the role of negative affect on IADLs in community-dwelling older adults with no depression: https://doi.org/10.1016/j.jadr.2022.100391).
Response 7: Thank you for your very thought-provoking opinion on depressive symptomatology and negative affect in the elderly. I read your paper and learned that negative affect (NA) is associated with financial capability even in the absence of depression. I believe that depression and negative affect have a significant impact on the overall life of the elderly, as indicated in the introduction to the paper you referred us to indicates. However, we did not assess depression and negative affect in the subjects in this study, so we added this as a limitation of this study by citing the paper you referred to P10, L478-481.
Comment 8: Please describe in more detail the recruitment procedure.
Response 8: Thank you for your comments on the recruitment procedure. We have added some information about the recruitment procedure in Method 2.3 Participants (P3, L126-129, L133-136), so please check it out.
Comment 9: Was MMSE administered? If no, why? Please justify as most studies use this measure for the assessment of global cognitive functioning.
Response 9: Thank you for your comments about the cognitive function test. As you point out, the MMSE should originally be administered to assess subjects' cognitive function. However, in our study, we were not able to do so due to the limitations of the physical environment of the location where the assessment was conducted and the difficulty in securing experts to assess the cognitive function tests. This is newly added as a limitation of this study (P10, L477-478). It is confirmed that the subjects in this study were not diagnosed with dementia by a doctor. Furthermore, another rationale is that the subjects in the analysis were not certified as needing long-term care under the Japanese long-term care insurance system (P3, L129-133).
Comment 10: Effect sizes are missing. Please provide them for all applied statistics.
Please follow APA guidelines for presenting statistics in the main text.
Response 10: Thank you for your comments regarding the description of the statistical analysis results. Please confirm that we have added the effect size to the statistical results (Table 1 - 3). Also, the coefficient of determination (X2), which is the effect size for multiple regression analysis, is noted in the text.
Comment 11: There are too many tables. Authors should think about using figures.
Response 11: Thank you for your remarks about the use of figures. We tried to show in the figures about Table2, 3 or about the results of the multiple regression analysis based on your suggestion, but the number of figures became too large. We understand that figures are easier to understand visually. However, for the results of this study, we decided that a table would more effectively convey the details of the results. We would appreciate it if you could understand our decision.
Comment 12: The Conclusions section is very brief. Please provide a more detailed section.
Response 12: Thanks for your comments on the conclusion. Please confirm that we have added the information to the conclusion, as other reviewers have advised us to do so (P10, L491-495).
Round 2
Reviewer 2 Report
Comments and Suggestions for Authors
It is ok in this revised version.